# Fish Consumption at One Year of Age Reduces the Risk of Eczema, Asthma and Wheeze at Six Years of Age

**DOI:** 10.3390/nu11091969

**Published:** 2019-08-21

**Authors:** Torbjørn Øien, Astrid Schjelvaag, Ola Storrø, Roar Johnsen, Melanie Rae Simpson

**Affiliations:** 1Department of Public Health and Nursing, NTNU—Norwegian University of Science and Technology, 7030 Trondheim, Norway; 2Frøya General Practice (Frøya legekontor), 7260 Sistranda, Norway; 3Clinic of Laboratory Medicine, St Olavs Hospital, 7030 Trondheim, Norway

**Keywords:** allergy, eczema, asthma, allergic rhinoconjunctivitis, fish, pregnancy, breastfeeding, diet

## Abstract

Background: The role of dietary fish and *n-*3 polyunsaturated fatty acids (*n*-PUFAs) in the primary prevention of allergic diseases remains uncertain. The aim of this study was to investigate associations between the consumption of fish and cod liver oil (rich in *n*-PUFAs) from pregnancy to the first two years of life, and parental reported allergic diseases at six years of age. Methods: We used data from the Prevention of Allergy among Children in Trondheim study and included mother-infant pairs who had submitted questionnaires detailing both maternal or infant diet and allergic disease at six years of age. Results: Eating fish at least once a week at one year of age was associated with a 28%, 40% and 34% reduction in the odds of current eczema, asthma, and wheeze at six years of age, respectively. Cod liver oil consumption at least four times per week at one year of age tended to be associated with a lower risk of allergy-related outcomes at six years. We found no consistent associations between allergy-related outcomes and fish or cod liver oil consumption by mothers. Conclusion: The preventive effect of fish consumption is best achieved by increasing dietary fish in the first year of life.

## 1. Introduction

Since the 1950s, we have witnessed a dramatic increase in the prevalence of allergy-related diseases, such as eczema (atopic dermatitis), asthma, allergic rhinoconjunctivitis (ARC, hay fever) and food allergies [1]. At the same time, there exists a large variability in the prevalence and severity of these diseases, not just between countries, but also across regions within the same country. This variability suggests environmental characteristics, such as eating habits, may influence the local prevalence [2]. Changes in dietary habits have also been flagged as one of the potential drivers of the increasing prevalence of allergy-related diseases. In particular, higher consumption of omega-6 polyunsaturated fatty acids (*n-*6 PUFAs) and lower consumption of omega-3 fatty acids (*n-*3 PUFAs) has been highlighted as one of the major changes in the Western diets which has occurred over the same time period as allergy-related disease prevalence has risen [3]. Plausible biologically mechanisms can explain a causal relationship between increased dietary *n-*6 PUFAS and allergy-related diseases [4]. *n-*3 PUFAs, on the other hand, are thought to have beneficial health effects [5] and both randomized controlled trials (RCTs) and epidemiological studies have sought to determine if higher dietary *n-*3 PUFAs prevents the development of allergy-related diseases by regulating the perinatal immune system [6,7]. The two largest RCTs assessing prenatal *n-*3 PUFAs supplementation have produced inconsistent results [8,9]. Bisgaard et al. observed a protective effect against persistent wheeze or asthma and lower respiratory tract infections up to five years [9], whilst Best et al. found protective effects only for atopic eczema and allergic sensitization up to six years [8]. A systematic review and meta-analysis of both randomized and quasi-randomized interventions concluded that maternal *n-*3 PUFA supplementation may be beneficial in the prevention of egg sensitization, but not other allergic outcomes [10]. Similarly, systematic reviews which have assessed trials of infant supplementation have failed to identify consistent preventive effects [11,12,13].

Oily fish is a major dietary source of *n-3* PUFAs, and several epidemiological studies have shown protective effects from eating fish during both pregnancy [14,15,16,17] and infancy [18,19,20] on the development of allergy-related diseases in childhood, however the results are conflicting. A small RCT, “The Salmon Pregnancy Study”, demonstrated salmon consumption in pregnancy modified neonatal immune responses, but had no effect on markers of infant atopy at six months of age [21]. Looking at observational studies, a systematic review and meta-analysis found no association between fish intake during pregnancy and allergy in the offspring, however fish consumption in the first year of life was associated with reduced risk of eczema and allergic rhinitis [22]. Another meta-analysis concluded that introduction of fish early in life (six to nine months) and intake of fish at least once a week reduces asthma and wheeze in children up to 4.5 years of age [23]. Few publications have considered maternal and infant fish or cod liver oil consumption within the same study and, due correlations between maternal and infant dietary habits, there is a need for studies considering both in order to disentangle their effects. 

We have previously published results from the Prevention of Allergy among Children in Trondheim (PACT) study describing a 30% reduction in doctor-diagnosed asthma at two years of age after a community-based lifestyle intervention designed to increase fish and cod liver oil intake, reduce tobacco exposure and reduce indoor dampness during pregnancy and the first two years of life [24]. From the same study, we also found a negative association between eating fish once a week at one year and eczema at two years [25]. 

The aim of the current study was to extend our previous work on diet and allergy-related disease in infancy and investigate associations between maternal and infant consumption of fish and cod liver oil on allergy-related diseases at six years of age. Using the PACT study, we sought to estimate the effect of four exposures (consumption of any, oily or lean fish, or cod liver oil), at four time points (pregnancy and breastfeeding for mothers and one- and two-years for infants) on four allergy-related diseases/symptom at six years of age (current eczema, asthma and wheeze, and cumulative incidence of ARC). 

## 2. Materials and Methods 

### 2.1. Study Population 

Data was obtained from the PACT study [26]. The PACT study is a controlled, population-based, primary intervention which aimed to reduce the incidence of allergy-related diseases through a multifaceted lifestyle intervention during pregnancy and the first two years of life. This municipality-wide lifestyle intervention involved structured advice regarding: Increased dietary intake of oily fish and cod liver oil, reduced parental smoking, and reduced indoor dampness. The design of the PACT study and recruitment to the “control” and “intervention” cohorts are further detailed in the Appendix A. Since the aim of the present study is to estimate associations between dietary fish or cod liver oil and allergic outcomes, rather than the effect of the intervention per se, we have included data from both the “intervention” and “control” cohorts and have adjusted for cohort membership in all statistical analyses. 

In total, 20,544 infants were recruited from routine prenatal and child health follow-ups at general practices and community health centers throughout Trondheim municipality, Norway. Sociodemographic characteristics, relevant risk factors, and dietary and smoking status were obtained from up to four lifestyle questionnaires completed once during pregnancy and when the child was six weeks, one-year and two-years-old. In addition, child health questionnaires, focusing primarily on signs and symptoms of allergy-related disease, were completed when the child was two and six years old. The inclusion period for all age groups commenced in September 2000 and ended in 2006 for pregnant women, with ongoing inclusion of one-, two- and six-year-olds until 2008, 2009, and 2014, respectively. Due to the study design, not all families were followed from the prenatal period, however more than half of the participants completed at least one of the lifestyle questionnaires. In the current analyses, we have included all families who provided information about maternal or infant diet in at least one lifestyle questionnaire and submitted the child health questionnaire at six years of age (Figure 1). The number of families varies for each analysis depending on the availability of exposure, outcome, and covariates.

Although it is not the focus of the current analyses, the PACT intervention did include advice about dietary intake of fish and cod liver oil. Specifically, women were advised to eat oily fish as dinner twice a week during pregnancy and take 5 mL of cod liver oil daily (equal to 1.2 g *n-*3 PUFA). Their children were advised to start with 5 mL cod liver oil from four to six weeks of age, and with oily fish as dinner or sandwich spread from six months of age. Cod liver oil is a common and traditional dietary supplement in Norway, which has been used over many generations, however not all pregnant women and infants take this supplement. 

### 2.2. Exposure Variables: Dietary Fish and Cod Liver Oil 

Dietary information was collected using a set of validated semiquantitative food-frequency questions in each of the lifestyle questionnaires. Specifically, the consumption of lean fish (e.g., cod, pollock/coalfish) and oily fish (e.g., ocean perch, halibut, salmon, trout, herring, and mackerel) for dinner were reported in two separate questions using the following frequency categories: never, less than once per week, once per week, twice per week, three times per week, and four or more times per week [24,25]. The frequency of cod liver oil (or fish oil capsule) supplementation was also categorized using this scale. Additionally, the consumption of oily fish as a sandwich spread was categorized into seven frequencies: Never, less than one slice per week, one to two slices per week, three to six slices per week, one to two slices per day, three to four slices per day or five or more slices per day. The same questions and frequency scales were used for maternal and infant dietary questions. 

Participating mothers were asked how often they ate fish or consumed cod liver oil when they were pregnant in three questionnaires: Pregnancy, six weeks and one year postpartum. Similarly, maternal fish and cod liver oil consumption whilst breastfeeding was recorded at both six weeks and one year postpartum (Figure 1). In cases where maternal dietary information was available from two or more questionnaires, the answer provided in the first/earlier questionnaire was preferentially used in further analyses to reduce the risk of recall bias. Infant dietary fish and cod liver oil information was available in the one- and two-year questionnaires. In addition to the food frequency questions, the one- and two-year questionnaires also included a question about the age at which the infant first ate fish of any sort. The introduction of fish was preferentially used from the one-year questionnaire when available. 

### 2.3. Outcome Variable: Eczema, Asthma, Wheeze and Allergic Rhinoconjunctivitis 

The presence of current eczema, current asthma, current wheeze and ever allergic rhinoconjunctivitis (ARC) were assessed from the six-year child health questionnaire. The questions were translated from the International Study of Asthma and Allergies in Childhood (ISAAC) questionnaire [27] adapted for the age group and tested for reliability [28]. Current eczema was defined using three questions in combination: ‘Has the child ever had eczema?’ and ‘Has the child ever had an itchy rash which came and went over at least six months?’ and ‘Has the child during the last 12 months used medications, balms, creams, tablets or herbal medicines for eczema?’. Current asthma was defined using the following two questions: ‘Has the child ever been diagnosed with asthma by a doctor?’ and ‘Has the child been treated with tablets, inhalation medications or other treatments for wheezing, tightness in the chest or asthma during the last 12 months?’. Current wheeze was defined using three questions “Has the child ever had whistling in the chest?” and “Has the child ever had episodes of wheezing or tightness in the chest” and “Has the child had whistling, rattling or tightness in the chest in the past 12 months”. Lastly, ARC was defined by ‘Has the child ever had hay fever, sneezing or itchy-watery eyes?’. The age of onset of the different outcomes was also recorded to account for reverse causality. 

### 2.4. Covariates 

Family history of allergy, the presence of older siblings, parental smoking during pregnancy, and the first year of a child’s life, breastfeeding duration, socioeconomic status and PACT cohort membership (control or intervention) were identified a priori as potential confounders of associations between maternal/infant diet, and allergy-related disease. Family history of atopy and presence of siblings were preferentially obtained from the pregnancy and six-week questionnaires and supplemented with information from the one-year questionnaire as required. Maternal and paternal cigarette smoking status was recorded in each of the lifestyle questionnaires and parental smoking was dichotomized based on whether either parent had reported smoking in any of the pregnancy, six-week, or one-year questionnaires. Breastfeeding duration was dichotomized as “any breastfeeding for six months or longer” and “weaning before six months” based on information in the one-year and two-year questionnaires. Maternal age and the average income of the family’s residential postal code were included as markers of socioeconomic status. The latter was obtained from summarized data from Norwegian tax lists from 2009 [29].

### 2.5. Statistical Analysis 

Statistical analyses were performed using Stata/MP 15.1 (StataCorp, College Station, TX, USA). A two-sided significant level of 0.05 is used to describe statistically significant findings. Non-statistically significant findings observed to have large effect sizes are also highlighted. 

#### 2.5.1. Description of Population and Dietary Correlations 

Differences between included subjects and drop-outs were assessed using Chi2 test for binary characteristics and Students’ *t*-test for continuous characteristics. Correlations between reported diet during pregnancy, breastfeeding, one-year and two-years was presented using Kendall’s tau-b and asymptotic standard error (ASE), agreement and relative risk (RR). 

#### 2.5.2. Main Analyses 

The analyses presented in this paper consider the effect of four exposures (consumption of any, oily or lean fish, or cod liver oil), at four time points (pregnancy and breastfeeding for mothers and one- and two-years for infants) on four allergy-related diseases/symptoms at six years of age (eczema, asthma, wheeze and ARC). The consumption of oily and lean fish was dichotomized into “one or more times per week” and “less than once a week”. Eating three to six slices of oily fish per week was considered equivalent to eating one meal of oily fish per week, such that mothers and infants who reported eating three to six or more slices with oily fish spread per week were also categorized as eating oily fish “one or more times per week”. Consumption of “any fish” was determined from the dichotomized variables for oily fish and lean fish. Cod liver oil supplementation was categorized into three groups: “Never”, “one to three times per week” or “more than four times per week”. Each combination of exposure, time point, and disease was modeled using logistic regression to obtain a crude odds ratio (OR) and adjusted OR (aOR) which had been controlled for the previously described covariates: Family history of allergy, presence of older siblings, parental smoking during pregnancy or first year, maternal age at birth, average income of residential postcode, and PACT cohort. Subsequently, each analysis was stratified by infant sex to assess the possibility that sex modified the effect of the fish or cod liver oil consumption on allergy-related diseases.

The association between age of introduction of fish into the infant’s diet and later allergy-related disease was also assessed using crude and adjusted logistic regression models for each allergy-related disease. For these analyses, age of fish introduction was categorized into six groups: “Before six months”, “6–9 months”, “9–12 months”, “12–15 months”, “after 15 months” and “not yet introduced (by two years)”. Exploratory investigations showed an approximately linear relationship between the prevalence of allergy-related diseases and these categories, such that the age of fish introduction was included as a continuous variable in the logistic regression models. 

#### 2.5.3. Sensitivity Analyses

A number of sensitivity analyses were conducted to assess the robustness of the results. First, an alternate multivariable logistic regression model (model 2) was conducted for each comparison to assess the influence of other dietary variables on the estimated associations. Specifically, these alternate models included dietary intake of the relevant fish/supplement at other time points and the consumption at the same time point of either cod liver oil (for models with fish intake as exposure) or any fish (for models with cod liver oil as exposure). Breastfeeding at six months of age was also included as a potential confounder for models examining the association between infant diet, and allergy-related disease. Models are described in the footnotes of each table. 

Second, the possible influence of recall bias in the reporting of maternal diet during pregnancy was assessed in the subset of mother-infant pairs with the information provided in the pregnancy questionnaire, i.e., excluding mother-infants pairs with diet in pregnancy information only available from the six-week or one-year questionnaires. Similarly, the influence of recall bias in reporting of diet whilst breastfeeding was assessed using the subset of mother-infant pairs with information from the six-week questionnaire.

Third, the influence of reverse causality was investigated by excluding infants who had either doctor-diagnosed asthma, parentally reported eczema or parentally reported ARC before (a) six months and (b) one year of age. Age of symptom debut or diagnosis was obtained from the two-year health questionnaire and supplemented with information from the six-year health questionnaire if no two-year questionnaire was available. 

### 2.6. Ethics Approval

The study was approved by The Regional Committee for Medical Research Ethics for Central Norway (Ref 120-2000), and the study was granted license by the Norwegian Data Inspectorate to process personal health data (Ref 2003/953-3 KBE/-). One parent signed a written informed consent on behalf of their child. 

## 3. Results

### 3.1. Participant Characteristics 

In total, 10,436 participants had provided information on at least one of the exposure variables, of which 4264 had also completed the child health questionnaire and could be included in at least one of the present analyses (Figure 1). These participants had similar baseline characteristics to those who completed at least one lifestyle questionnaire but dropped out of the study before the six-year follow-up (Table 1). Compared to drop-outs, included families were more likely to have older siblings and breastfeed beyond six months, the mothers were marginally older and both mothers and fathers were less likely to have smoked during pregnancy and the first year of life. These families also tended to live in postcodes with higher average incomes (Table 1). For each exposure time point, a different set of participants could be included and considered as drop-outs. However, the differences in characteristics were relatively constant across exposure time points (available in Appendix A). For women recruited during pregnancy, the median gestational age at completion of the pregnancy questionnaire was median 13 weeks (IRQ 10–21).

### 3.2. Maternal and Infant Dietary Fish and Cod Liver Oil Intake: Description and Correlations 

Regular fish consumption was common in this population. Eating any kind of fish at least once per week was reported by 63% of women during pregnancy and 65%, whilst breastfeeding, and by 51% and 72% of infants at one and two years of age, respectively. The consumption of lean fish was more common at all exposure time points. For cod liver oil, approximately 35–40% of mother and infants reported “never” consuming cod liver oil, whilst 35–45% consumed cod liver oil more than four times per week (further details found in Appendix A). 

There is a strong correlation between maternal fish consumption during pregnancy and whilst breastfeeding (Kendall τ-b = 0.63–0.66, Appendix A) and a weak to moderate correlation between maternal and infant consumption (Kendall τ-b = 0.25–0.36, Appendix A). Subsequently, the majority of mother–infant pairs fell into three broad categories: Neither consumed fish (18%), only the mother consumed fish and did so both in pregnancy and whilst breastfeeding (21%), or fish was consumed by both mother during pregnancy and breastfeeding, and the child at one year (36%) (Appendix A). We, therefore, had limited opportunity to consider the effect of fish consumption in pregnancy only or at one year alone. 

### 3.3. Maternal Fish and Cod Liver Oil Intake During Pregnancy and Allergy-Related Outcomes at Six Years 

Maternal intake of any fish during pregnancy tended to be associated with a higher risk of ARC with an aOR of 1.24 (95% CI 0.97–1.60), and oily fish tended to be associated with a higher risk of asthma, wheeze, and ARC with aORs of 1.34 (95% CI 0.94–1.92), 1.27 (95% CI 1.00–1.63) and 1.23 (95% CI 0.97–1.56), respectively (Appendix A). None of these observed associations reached statistical significance. After controlling for other dietary factors, the positive association between any or oily fish intake were stronger for both asthma and ARC (model 2, Appendix A). While this would suggest that the direct effect of fish consumption in pregnancy may not be beneficial, we could not reliably review these findings in specific subgroups due to the small proportions of mother-infant pairs “exposed” to any or oily fish only during pregnancy. 

Cod liver oil consumption during pregnancy was not statistically associated with eczema, wheeze or ARC (Appendix A). Consumption of cod liver oil one to three times per week during pregnancy was associated with a reduction in the risk of current asthma at six years of age (aOR 0.57, 95% CI 0.32–1.01), however this preventive effect was not apparent for consumption of cod liver oil four or more times per week (aOR 0.96, 95% CI 0.65–1.42). None of the estimated associations between cod liver oil in pregnancy and allergy-related diseases changed substantially after adjusting for other dietary factors (model 2 in Appendix A). 

There was no clear indication that infant sex modified any associations between maternal fish intake during pregnancy and allergy-related disease (Appendix A). Cod liver oil intake more than four times per week in pregnancy was associated with higher prevalence of eczema, asthma, and ARC among girls compared to boys (Appendix A), however the inclusion of sex as an interaction term in the logistic regression models indicated that these differences were not statistically significant (data not shown). The influence of recall bias appears to be minimal as the effect estimates for fish and cod liver oil intake during pregnancy were largely unchanged in the subgroup with questionnaire data available from pregnancy (Appendix A). 

### 3.4. Maternal Fish and Cod Liver Oil Intake Whilst Breastfeeding and Allergy-Related Outcomes at Six Years 

Neither fish nor cod liver oil consumption, whilst breastfeeding was statistically associated with any of the allergy-related outcomes in crude or adjusted models (Appendix A). As for pregnancy, cod liver oil consumption during pregnancy appeared to be associated with a higher prevalence of eczema, asthma, and wheeze among girls compared to boys, but the differences did not represent statistically significant effect modification (Appendix A). The association between maternal diet while breastfeeding does not appear to be substantially affected by recall bias (Appendix A) but may have been influenced by reverse causality or confounding-by-indication (Appendix A). Specifically, after excluding infants with a diagnosis or symptoms of eczema, asthma or ARC before 6 or 12 months of age, we found that fish consumption, while breastfeeding (any, oily or lean) tended to be associated with a higher prevalence of current wheeze and ARC at six years. This observation was only statistically significant for oily fish consumption and current wheeze (aOR 1.52, 95% CI 1.08–2.14, Appendix A). 

### 3.5. Fish and Cod Liver Consumption by Infants at One Year and Two Years and Allergy-Related Outcomes at Six Years

Infant dietary intake of any fish or lean fish at one year of age was associated with a clinically and statistically significantly lower prevalence of current eczema, asthma, and wheeze at six years of age (Table 2). Similarly, the consumption of oily fish at one year of age was associated with a reduction in current asthma and wheeze, although these estimates did not reach statistical significance. The protective effects of infant fish intake at one year on these allergic outcomes remained after adjusting for maternal diet, breastfeeding at six months and infant cod liver oil consumption (model 2 in Appendix A). Furthermore, they did not appear to be modified by infant sex (Appendix A) and were largely unchanged after excluding infants who had developed symptoms of allergy-related disease before six months and one year of age (Appendix A). 

In an alternative regression model, we assessed the effect of consumption at least once a week of “oily fish only”, “lean fish only” and “both fish” compared to “no fish”. The consumption of lean fish only was associated with substantial reductions in current eczema, asthma, and wheeze at six years (Table 3). The consumption of oily fish alone or of both fish types was also estimated to have protective effects on these allergic outcomes, however, these estimates were mostly weaker, less precise and not statistically significant. The one exception was a marginally stronger protective effect against asthma observed among one-year-old children who consumed both fish types at least once per week (Table 3).

At one year of age, consumption of cod liver oil more than four times per week was also associated with a lower prevalence of current eczema, asthma and wheeze, although this was only statistically significant for current eczema (aOR 0.71, 95% CI 0.53–0.95). Once again, the preventive effect remained after adjusting for other dietary factors (Appendix A), stratifying by sex (Appendix A) and excluding symptomatic infants (Appendix A). The cumulative incidence of ARC was not associated with either fish or cod liver oil intake at one year of age. 

Consumption of fish and cod liver oil at two years of age was not associated with allergy-related outcomes at six years except for fish consumption and current wheeze. Consumption of both oily and lean fish was associated with a higher prevalence of current wheeze, aORs 1.37 (95% CI 1.01–1.84) and 1.41 (95% CI 1.03–1.95), respectively (Appendix A). Upon stratification by infant sex, only girls consuming fish at least once per week at two years of age had increased prevalence of current wheeze (Appendix A). The effect of any and oily fish consumption at two years of age on current asthma at six years also appeared to be modified by sex with a slightly protective effect observed for boys and a higher risk seen for girls (Appendix A). Excluding infants with symptoms before six months and one year of age resulted in a stronger positive association between fish consumption at two years and current wheeze at six years (Appendix A). 

### 3.6. Age of Introduction of Fish into Infant Diet and Allergy-Related Outcomes at Six Years 

Early introduction to fish was associated with a lower prevalence of each of the allergy-related outcomes in crude analyses (Table 4, Figure 2). After adjustment for potential confounders, the effect estimates were largely unchanged, however the smaller number of participants with complete information on the covariates means that these estimates are less precise and no longer statistically significant except for ARC. 

After excluding infants who had developed symptoms of eczema, asthma or ARC before six-months, the effect of early introduction to fish was substantially weakened for current wheeze and ever ARC (Table 4). We also found evidence of probable reverse causation or confounding-by-indication for all allergic outcomes when excluding infants with symptoms before one year of age. This sub-group analysis indicated that the age of introduction was not associated with allergic disease at six years (Figure 2, Table 4). 

## 4. Discussion

Fish consumption at least once a week at one year of age was found to be associated with a 28%, 40% and 34% reduction in the odds of current eczema, asthma, and wheeze at six years of age, respectively. These findings are consistent with our previous analyses indicating that infant fish consumption at one year reduced the risk of eczema at two years of age [25]. The protective effect of fish consumption at one year appears to be relevant for both boys and girls, is not conclusively associated with a particular fish type, is independent of maternal consumption during pregnancy and breastfeeding, and does not appear to be the result of reverse causality or confounding-by-indication. We also found that cod liver oil consumption at least four times per week at one year of age tended to be associated with a lower risk of allergy-related outcomes at six years. Otherwise, we found no other consistent associations between allergy-related outcomes and fish or cod liver oil consumption by mothers during pregnancy and breastfeeding, or infants at two years of age. Nor could we exclude that the apparent protective effect of early fish introduction is primarily a result of reverse causality or confounding-by-indication. 

These findings are in line with recent meta-analyses which found that infant consumption of fish, but not maternal consumption during pregnancy, was associated with a reduced risk of eczema and ARC and possibly asthma [22]. Similarly, a pooled analysis of multiple birth cohorts found that the risk of wheeze, asthma, and ARC in preschool and school-age children was not associated with maternal consumption of fish during pregnancy [30]. Together with the findings presented in this paper, these studies suggest that any preventive effect of fish consumption is best achieved by increasing fish in the infants’ diet. Indeed, our findings suggest that maternal fish consumption alone may tend to increase the risk of asthma, wheeze, and ARC. 

In an attempt to tease out the effect of maternal and infant fish consumption, we applied an alternative adjusted logistic regression model to each comparison, which included maternal or infant dietary consumption at other time points. As one would expect, we observed that maternal and infant consumption are correlated. Assuming that maternal consumption partially determines infant consumption at one year, when the association of maternal fish consumption during pregnancy and allergic outcomes is adjusted for infant’s consumption at one year of age we are essentially conditioning on a mediator. In doing so, we are no longer estimating the total effect of consumption during pregnancy but are essentially estimating the natural direct effect of fish consumption in pregnancy on allergic outcomes at six years. The estimates produced by model 2 suggesting that maternal fish consumption during pregnancy may increase the risk of asthma, wheeze and ARC are, therefore, not directly comparable to previous studies, and do not represent conflicting results. However, these findings need to be investigated in other large studies and considered in light of other health benefits of fish consumption for pregnant women. 

In a study published by Goksör et al. [31], the introduction of fish into the infants’ diet before nine months of age was associated with a reduced risk of wheeze at 4.5 years after adjusting for a number of potential confounders, including eczema and food allergy during the first year of life, and for frequency of fish consumption at one year of age. In contrast, our own findings suggest that the apparent association between early introduction to fish and lower risk of the allergic outcomes was probably influenced by delayed introduction of fish after the development of eczema, asthma or ARC. The discrepancy between our findings and those of Goksör et al. [31] findings may be due to differences in the measure of “frequent” fish consumption (at least once per month vs. per week), the choice of potential confounding factors, or the inclusion of fish consumption at one year as a covariate since this may partially mediate the effect of early introduction of fish on wheeze. 

Early introduction of allergenic foods, including fish, has more recently been investigated in the primary prevention of food allergy. A recent review found evidence that the immunoregulatory and anti-inflammatory properties of allergenic foods can provoke oral tolerance if introduced early to both low-risk and high-risk infants [32], and a number of RCTs have produced promising results for the early introduction of peanut and egg in the prevention of allergy to these foods in high-risk infants [33]. The benefits of early fish introduction as a means of preventing fish allergy are unknown. To date, only the EAT study has included fish in the early introduction regime [34], however, the incidence of fish allergy in the study participants was too low to be able to confirm whether the early introduction can prevent fish allergy. We did not have sufficient information about food allergy among the children in the PACT study to be able to investigate this possibility in the present study. 

In terms of cod liver oil consumption, our findings are in contrast with the results from the RCT by Bisgaard et al. [9] which found a reduced risk of recurrent wheeze and asthma up to five years of age after *n-3* PUFA supplementation during pregnancy. The supplement recommended in the PACT study was equivalent to 1.2 g *n-3* PUFA per day, half the dose used their RCT, so it is unclear if the contrasting effects are due to the dosage, residual confounding or misclassification in our study. It is also worth noting that the preventive effect reported by Bisgaard et al. [9] was not observed in the other large RCT [8] or meta-analyses [10] and needs to be confirmed in future studies. Another interesting observation made by Bisgaard et al. was that the preventive effect of *n-3* PUFA supplementation was most pronounced for infants of women with a low baseline eicosapentaenoic acid (EPA) and docosahexaenoic acid (DHA) levels. The PACT study design does not allow us to investigate this further, although it would suggest that encouraging *n*-PUFA intake in populations with normally high oily fish and cod liver oil consumption may only provide a small preventive benefit. In addition to baseline levels of PUFAs, other factors may influence the effect of *n-3* PUFAs on allergic outcome, such as vitamin D status or concomitant vitamin D supplementation. Results from the Vitamin D Antenatal Asthma Reduction Trial (VDAART) suggest that infants with high cord blood 25 (OH)D, in combination with high plasma or dietary PUFA at three years of age, was associated with the lowest odds of allergic disease [35]. This would suggest that antenatal vitamin D status may influence the magnitude of the preventive effect of postnatal PUFA consumption. Interaction between maternal vitamin D and PUFA supplementation was not observed by Bisgaard et al. [9]. In a separate analysis from the VDAART study, prenatal fish oil supplementation was associated with a reduced risk of asthma/recurrent wheeze in offspring, however, baseline vitamin D in mothers did not affect this preventive effect [36]. 

The strengths of this study are the large size of the study and that it comprises exposure both during pregnancy, lactation and in the child’s two years of life. Information on diet at each of these time points allows us to distinguish between the influence of the mother’s and child’s dietary consumption. The prospective design reduces the risk of recall bias, and we were able to consider the influence of reverse causality and confounding-by-indication by using parental reports of the age of disease or symptom onset. At the same time, limitations of this study include the risk of selection bias, the high drop-out rate, and the lack of objectively confirmed and repeated dietary intake information and allergic outcomes. The PACT study aimed to recruit a general population. To investigate the degree of self-selection bias, a non-participant study was conducted using a short questionnaire. Information regarding maternal age, allergic disease in the family, smoking and socioeconomic status was completed for 391 consecutive parents visiting maternal postnatal care centers. These non-participants were found to be similar to participants in these baseline characteristics [4]. For the current study, comparing participants from drop-outs indicated that there were some minor differences, however, we do not believe these have substantially influenced the results. Lastly, the lack of objectively confirmed dietary intake and parentally reported allergic outcomes may introduce a degree of misclassification of these measures, although we consider it most likely that any misclassification is non-differential. Only one questionnaire was completed during pregnancy and we, therefore, do not have the possibility to determine how representative the reported dietary habits are of the entire pregnancy period, nor can we use this data to investigate the importance of the timing of fish or cod liver oil consumption during pregnancy. Analyses from the VDAART study suggest that consumption during both the first and third trimester is of the greatest preventive effect against asthma/recurrent wheeze [36], however, this is in a population with comparatively very low fish oil consumption. Allergic outcomes were parentally reported at two and six years. While there was no specific follow-up of health outcomes among PACT participants before two years of age, we believe that most children with allergic symptoms would have been identified through routine follow-ups at community health centers. Throughout Norway, children attend free-of-charge routine follow-ups with a child health nurse at six weeks, and 3, 4, 5, 6, 8, 10, 12, 15, 18, and 24 months. Children are additionally assessed by a doctor at six weeks, and 6, 12, and 24 months. Nonetheless, the age of disease or symptom debut recorded in the PACT study may still be subject to recall bias. 

## 5. Conclusions

In line with previous meta-analyses, we found that the inclusion of fish in the infant diet by one year of age is more influential in the prevention of allergy-related disease than maternal consumption. Fish consumption at one year of age appears to reduce the risk of eczema, asthma, and wheeze at six years of age. The preventive effect of fish consumption could not be specifically attributed to either oily or lean fish consumption and the estimates were robust under sensitivity analyses. Similarly, taking cod liver oil at least four times per week was observed to have a protective effect, although these estimates did not reach statistical significance. On the other hand, neither maternal consumption of fish or cod liver oil during pregnancy or breastfeeding nor infant consumption at two years of age were associated with allergy-related disease at six years of age. 

## Figures and Tables

**Figure 1 nutrients-11-01969-f001:**
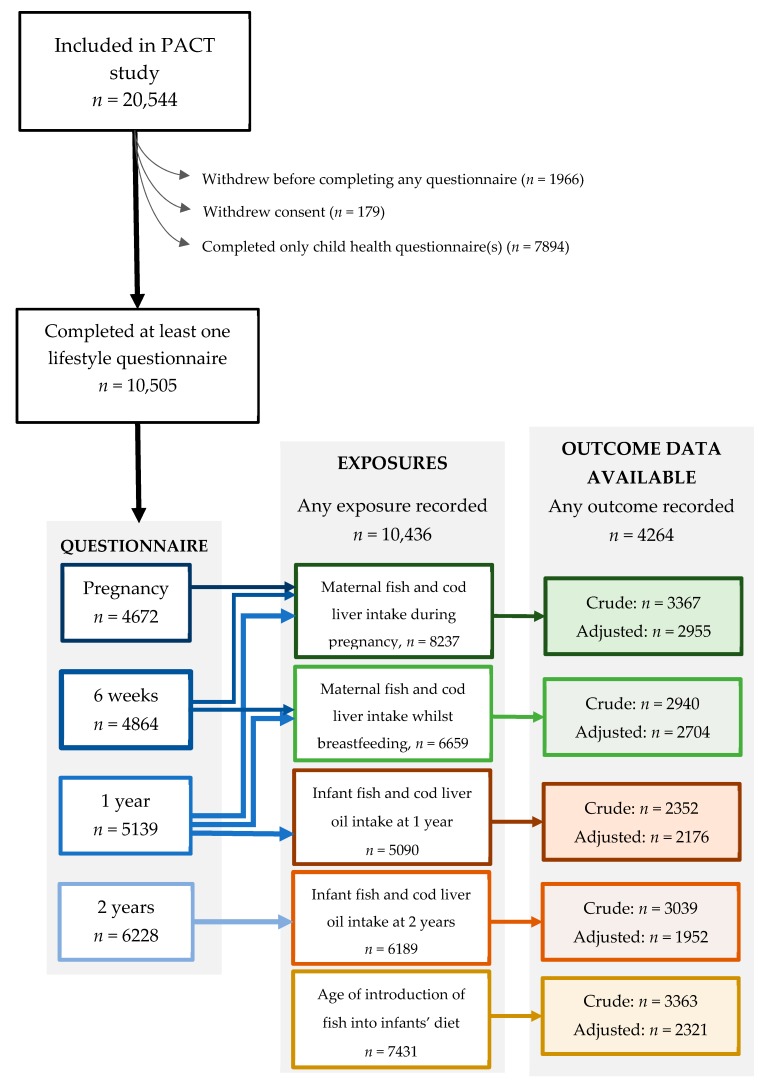
Flow chart for participants. The exact number of participants in each analysis differs slightly due to missing observations for individual exposures and outcomes.

**Figure 2 nutrients-11-01969-f002:**
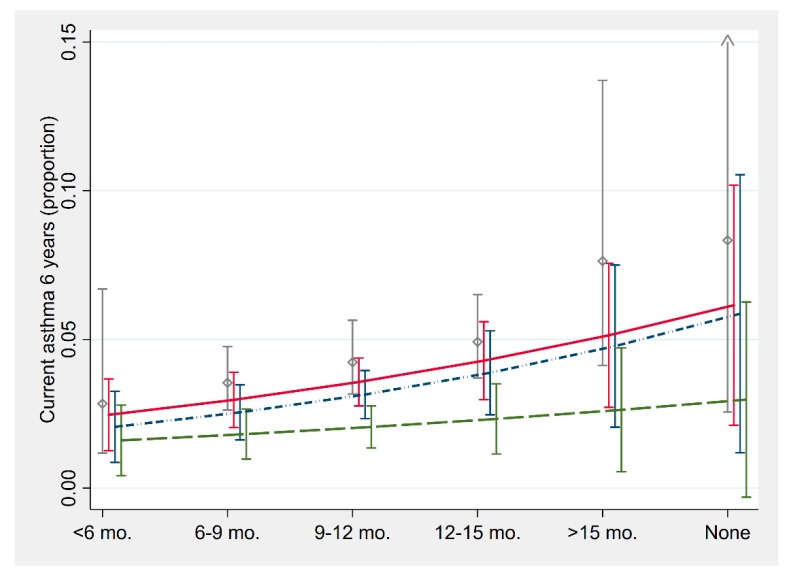
The proportion of children with current asthma at six years of age over the age fish was introduced into the infants’ diet. This graph presents the observed proportions (grey diamonds) and predicted proportions from adjusted logistic regression models including all participants (red solid line) and excluding infants with diagnosis or symptoms before six months (blue dashed/dotted line) and 12 months (green dashed line). Spikes indicate 95% CI. Delayed introduction of fish was associated with increased prevalence of asthma in all models, although after removing infants who had developed allergic symptoms before 12 months of age the association between age of introduction and prevalence of asthma is substantially weakened (green dashed line).

**Table 1 nutrients-11-01969-t001:** Baseline characteristics of included mother-infant pairs and drop-outs.

	Included (*N* = 4264)			Drop-outs (*N* = 6241)			
Binary Covariates	N	n	%	N	n	%	*p*-Value
Sex, male	4261	2131	50.0	5339	2677	50.1	0.900
Family history	3224	2264	70.2	4535	3332	73.5	0.002
Older siblings	3229	1823	56.5	4525	2323	51.3	<0.001
Breastfeeding at 6 mo	3702	3134	84.7	4442	3571	80.4	<0.001
**Mother smoking**							
Pregnancy	1896	130	6.9	2488	213	8.6	0.037
6 weeks	2178	165	7.6	2405	240	10.0	0.004
1 year	2319	348	15.0	2678	480	17.9	0.006
Ever in first year	3334	466	14.0	4793	752	15.7	0.033
**Father smoking**							
Pregnancy	1852	296	16.0	2395	481	20.1	0.001
6 weeks	2072	340	16.4	2299	451	19.6	0.006
1 year	2164	371	17.1	2490	506	20.3	0.006
Ever in first year	3230	665	20.6	4573	1059	23.2	0.007
**Parental smoking**							
Pregnancy	1830	316	17.3	2371	511	21.6	0.001
6 weeks	2040	367	18.0	2225	458	20.6	0.032
1 year	2144	460	21.5	2459	612	24.9	0.006
Ever in first year	3209	779	24.3	4516	1204	26.7	0.018
Cohort, intervention	4264	1204	28.2	6241	1714	27.5	0.385
**Continuous covariates**	**N**	**mean**	**SD**	**N**	**mean**	**SD**	
Maternal age, yrs	4244	30.3	4.4	5258	29.3	4.7	<0.001
Birthweight, gm	3944	3583	565	5100	3579	597	0.716
Breastfeeding duration, mo.	3392	10.7	5.6	3966	9.8	5.7	<0.001
Income, NOK	4158	257,043	28,563	5664	250,535	31,617	<0.001
Maternal education, yrs	2573	15.7	2.6	1331	15.5	2.6	0.051
Paternal education, yrs	2543	15.1	3.0	1314	15.1	3.0	0.896

**Table 2 nutrients-11-01969-t002:** Association between infant dietary fish at one year and allergic disease/symptoms at six years: Crude and adjusted models.

		≥1 Time/Week			<1 Time Per Week			Crude	Adjusted (Model 1)		
	N	N_1_	*n* _1_	%	N_0_	*n* _0_	%	OR	aOR	95% CI	*p*-Value
**Any fish**											
Current eczema	2125	1108	125	11.3	1017	153	15.0	0.73	0.72	0.56–0.93	0.011
Current asthma	2162	1129	37	3.3	1033	57	5.5	0.58	0.60	0.39–0.91	0.018
Current wheeze	2139	1115	93	8.3	1024	131	12.8	0.61	0.66	0.50–0.88	0.005
Ever ARC	2023	1067	121	11.3	956	118	12.3	0.93	0.92	0.70–1.21	0.534
**Oily fish**											
Current eczema	2126	579	74	12.8	1547	204	13.2	1.03	0.95	0.71–1.26	0.713
Current asthma	2163	590	20	3.4	1573	74	4.7	0.71	0.72	0.43–1.20	0.211
Current wheeze	2140	583	51	8.7	1557	173	11.1	0.74	0.83	0.60–1.16	0.276
Ever ARC	2024	556	69	12.4	1468	170	11.6	1.11	1.07	0.79–1.45	0.657
**Lean fish**											
Current eczema	2129	966	110	11.4	1163	169	14.5	0.74	0.76	0.59–0.99	0.043
Current asthma	2166	985	30	3.0	1181	64	5.4	0.54	0.56	0.36–0.88	0.012
Current wheeze	2143	973	82	8.4	1170	143	12.2	0.65	0.69	0.52–0.92	0.012
Ever ARC	2027	932	101	10.8	1095	139	12.7	0.86	0.85	0.65–1.12	0.250

N: total number of participant included in the analysis; N_1_ and N_0_: number of participants eating the fish at least once per week or less than once per week, respectively; n_1_ and n_0_: number of participants with and without the allergic outcome in each group; ARC: Allergic rhinoconjunctivitis, aOR: Adjusted odds ratio from logistic regression analysis for each fish exposure and each condition, adjusted for family history of atopy, presence of older siblings, parental smoking within during pregnancy and the first year of life, maternal age, average income of postal code area and interventional cohort (Model 1).

**Table 3 nutrients-11-01969-t003:** Consumption of specific fish types at one year of age and allergy-related outcomes at six years of age.

		Oily Fish Only Vs. None			Lean Fish Only Vs. None			Both Fish Vs. None		
	N	aOR	95% CI	*p*-Value	aOR	95% CI	*p*-Value	aOR	95% CI	*p*-Value
Current eczema	2125	0.68	0.39−1.17	0.164	0.61	0.44−0.86	0.005	0.86	0.62−1.20	0.388
Current asthma	2162	0.87	0.38−1.96	0.731	0.58	0.33−1.00	0.051	0.53	0.29−0.99	0.047
Current wheeze	2139	0.69	0.42−1.29	0.243	0.61	0.42−0.88	0.008	0.73	0.50−1.06	0.100
Ever ARC	2023	1.25	0.75−2.08	0.396	0.83	0.58−1.17	0.279	0.93	0.65−1.33	0.678

ARC: Allergic rhinoconjunctivitis, aOR: Adjusted odds ratio from logistic regression analysis for each fish exposure and each condition, adjusted for family history of atopy, presence of older siblings, parental smoking within during pregnancy and the first year of life, maternal age, average income of postal code area and interventional cohort (Model 1). Number of infants in each fish consumption category varies depending on missingness in outcome variables: “Oily fish only” (*n* = 138–147), “lean fish only” (*n* = 511–539) and “both fish” (*n* = 418–443) compared to “no fish” (*n* = 956–1033).

**Table 4 nutrients-11-01969-t004:** Association between the age of introduction of fish and allergic disease at six years.

	CrudeOR ^‡^	Adjusted (Model 1)			Infants with Symptoms before 6 Months Excluded			INFANTS with Symptoms before 12 Months Excluded		
		N	aOR ^‡^	95% CI	N	aOR ^‡^	95% CI	N	aOR ^‡^	95% CI
Current eczema	0.88	2369	0.89	0.78−1.02	1987	0.90	0.76−1.06	1758	0.95	0.77−1.17
Current asthma	0.81	2406	0.83	0.66−1.03	2012	0.80	0.62−1.05	1777	0.88	0.62−1.25
Current wheeze	0.89	2382	0.93	0.80−1.07	1992	0.98	0.82−1.15	1760	1.03	0.85−1.25
ARC	0.85	2245	0.86	0.75−0.98	1903	0.96	0.81−1.14	1695	0.93	0.77−1.13

ARC: Allergic rhinoconjunctivitis, aOR: Adjusted odds ratio from logistic regression analysis for each allergic outcome adjusted for family history of atopy, presence of older siblings, parental smoking within during pregnancy and the first year of life, maternal age, the average income of postal code area and interventional cohort (Model 1). ^‡^ the ORs and aORs represent the change in odds of the allergic outcome from one age of introduction category to an earlier category (e.g., 12–15 mo. compared to 9−12 mo., or 9−12 mo. compared to 6−9 mo.)

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
