# Peer review of "Fish Consumption at One Year of Age Reduces the Risk of Eczema, Asthma and Wheeze at Six Years of Age"

_nutrients, 2019, doi:10.3390/nu11091969_

Round 1

Reviewer 1 Report

The authors present in the manuscript results about the effect that fish consumption during the first two years of life, along with the fish consumption of the mother during pregnancy and breastfeeding has on the development of allergic pathology at six years of age. The authors handled a large amount of data and demonstrated experience both on the subject of the manuscript and in the analysis of large quantities of results.

The introduction is very clear and the objective of the work is clearly defined.

The authors present in the results a large amount of data that in many cases are difficult to integrate and assimilate. The authors should make an effort to synthesize the results that are important to obtain the conclusions of the work. For example in point 3.2. Maternal and infant dietary fish and cod liver oil intake: description and correlations, there is an excess of results to conclude that of mothers who consumed fish during their pregnancy the children also consume it in their early stages of life.

In the section, 3.3. Maternal fish and cod liver oil intake while pregnant or breastfeeding and allergy-related outcomes at 6 years, contradictory results are presented. Initially it is indicated that the consumption of pesacado does not influence the development of allergy and later it is mentioned that if there is a tendency towards an increase of asthma and ARC. This aspect must be clarified based on the results obtained.

Both in table 2 and table 3 it is clearly reported that the protective effect of the fish is strongly associated with lean fish, leaving this effect much more diluted altrotar de oily fish. This data is somewhat lost among the rest and should be presented and discussed clearly.

Table 4 is very confusing. The relationship between the time of introduction of the fish in the diet and the effects on allergy at 6 years is not appreciated. It must be presented more clearly.

Author Response

Reviewer 1
The authors present in the manuscript results about the effect that fish consumption during the first two years of life, along with the fish consumption of the mother during pregnancy and breastfeeding has on the development of allergic pathology at six years of age. The authors handled a large amount of data and demonstrated experience both on the subject of the manuscript and in the analysis of large quantities of results.
The introduction is very clear and the objective of the work is clearly defined.
The authors present in the results a large amount of data that in many cases are difficult to integrate and assimilate. The authors should make an effort to synthesize the results that are important to obtain the conclusions of the work. For example in point 3.2. Maternal and infant dietary fish and cod liver oil intake: description and correlations, there is an excess of results to conclude that of mothers who consumed fish during their pregnancy the children also consume it in their early stages of life.
Thank you for the suggestion. We have revised the text throughout the results to make it more succinct. Section 3.2. has been particularly condensed. We have opted to keep a brief introductory paragraph describing the fish consumption habits of the mothers and infants in our study before presenting the correlations between mother and infant diets.
We have also updated the Supplementary Material so that it has a table of contents and subheadings. We hope this will help readers navigate the large quantities of results, whilst maintaining transparency in the reporting of results.
In the section, 3.3. Maternal fish and cod liver oil intake while pregnant or breastfeeding and allergy-related outcomes at 6 years, contradictory results are presented. Initially it is indicated that the consumption of pesacado does not influence the development of allergy and later it is mentioned that if there is a tendency towards an increase of asthma and ARC. This aspect must be clarified based on the results obtained.
Thank you for pointing out that this paragraph could be confusing. We have rearranged this paragraph to clarify these findings (Lines 253-257, in clean manuscript) :
“Maternal intake of any fish during pregnancy tended to be associated with a higher risk of ARC with an aOR of 1.24 (95% CI 0.97-1.60), and oily fish tended to be associated with a higher risk of asthma, wheeze and ARC with aORs of 1.34 (0.94-1.92), 1.27 (1.00-1.63) and 1.23 (0.97-1.56), respectively (Table S4). None of these observed associations reached statistical significance.”
Whilst we did not find statistically significant associations between fish consumption in
pregnancy and allergic outcomes, the observed effect estimates (aORs) and their confidence
intervals are close to being statistically significant.
Both in table 2 and table 3 it is clearly reported that the protective effect of the fish is strongly
associated with lean fish, leaving this effect much more diluted altrotar de oily fish. This data is
somewhat lost among the rest and should be presented and discussed clearly.
We have modified the text to more clearly described the greater certainty in the results for
lean (or ‘any’) fish by rephrasing part of these results (lines 305 – 312 in the clean
manuscript) :
In an alternative regression model, we assessed the effect of consumption at least
once a week of “oily fish only”, “lean fish only” and “both fish” compared to “no
fish”. The consumption of lean fish only was associated with substantial reductions
in current eczema, asthma and wheeze at 6 years (Table 3). The consumption of oily
fish alone or of both fish types were also estimated to have protective effects on
these allergic outcomes, however, these estimates were mostly weaker, less precise
and not statistically significant. The one exception was a marginally stronger
protective effect against asthma observed among 1-year-old children who consumed
both fish types at least once per week (Table 3).
We agree that the effect estimates (aOR) for lean fish demonstrate a stronger protective
effect, however in most cases the aOR differs only marginally from the aOR estimated for oily
fish. Nonetheless, estimates of the effect of oily fish consumption, and particularly oily fish
consumption alone (Table 3), have very low precision because there were substantially fewer
infants eating oily fish at 1 year of age (~ 7 % compared to 25 % eating lean fish alone). We
therefore observe wide confidence intervals and no statistically significant findings.
Nonetheless, we cannot dismiss the possibility that oily fish also having a protective effect
based on these results and we consider it important to highlight that this.
Table 4 is very confusing. The relationship between the time of introduction of the fish in the diet
and the effects on allergy at 6 years is not appreciated. It must be presented more clearly.
We have added a foot note in the Table 4 explaining how the ORs can be interpreted. We
have also modified Table 4 to include the analysis after excluding infants who had developed
symptoms before 1 year. This means that it is easier to appreciate the influence these infants
have on the results without consulting the Supplementary Material. Finally, we have
included predicted estimates for the prevalence of asthma in Figure 2. In particularly, this
graph visually depicts the weakened association between age of introduction and asthma at
6 years when we remove infants who had developed symptoms before 1 year of age (green
dashed line).

Reviewer 2 Report

This is a very intersting paper on the impact of dietary fish consumption and n-PUFAs at different time-points of early-life. 

My comments are : 

- I suggest not to include any results in the introduction (page 2, lines 68-73).

- Statistical tests used for the comparison between the included subjects and the drop-outs have not been mentioned in the paragraph "statistical analyses". 

- The discussion may be slightly enriched, such as with potential explanations of the reasons why no significant result was found for maternal fish consumption during pregnancy and during breastfeeding.

Author Response

Reviewer 2
This is a very interesting paper on the impact of dietary fish consumption and n-PUFAs at different
time-points of early-life.
My comments are :
- I suggest not to include any results in the introduction (page 2, lines 68-73).
We agree that this is unusual in many medical journals, however the template supplied at
Nutrients instructs authors to “… briefly mention the main aim of the work and highlight the
principal conclusions” at the end of the Introduction. It appears that other articles published
in this special addition have not included results in the introduction and we have therefore
deleted these lines in the revised manuscript. If the editors would prefer us to follow the
template provided by the journal, these lines can be reintroduced.
- Statistical tests used for the comparison between the included subjects and the drop-outs have not
been mentioned in the paragraph "statistical analyses".
Thank you for pointing out this was not clear. We have now included a brief description of
the statistical analyses comparing included subject and drop-outs (§2.5.1.), as well as a
column in Table 1 presenting the p-values for these statistical comparisons. This comment
also prompted us to clarify what statistical methods were used to identify correlations for
dietary intake at the different timepoints and we have now included this in the same section.
- The discussion may be slightly enriched, such as with potential explanations of the reasons why no
significant result was found for maternal fish consumption during pregnancy and during
breastfeeding.
As described in lines 392-396, a lack of statistically significant association between pregnancy
intake and allergic outcomes is consistent with those of the recent meta-analysis (Zhang et
al, ref 22) and the large pooled analysis from 18 European cohorts (Stratakis et al, ref 30).
Together, these suggest that fish intake or cod liver oil during pregnancy may not necessarily
affect infant allergic outcomes. Perhaps this is because the any beneficial components are
not transferred in sufficient quantities over the placenta or perhaps these components have
little effect whilst the infant is a foetus and are of greater influence after birth. However, this
is speculative and we do not feel we have enough evidence to describe this further. Also, we
are aware that one of the larger RCTs employing prenatal supplementation found a
preventive effect of n-3 PUFAs in pregnancy on recurrent wheeze / asthma (Bisgaard et al, ref
9) and that the most recent meta-analysis on pregnancy supplementation with n-3 PUFA
suggest prevention of egg sensitization (Garcia-Larsen et al, ref 10). So perhaps this is due to
residual confounding and misclassification, discussed briefly in lines 421-427:
In terms of cod liver oil consumption, our findings are in contrast to the results from
the RCT by Bisgaard et al[9] which found a reduced risk of recurrent wheeze and
asthma up to 5 years of age after n-3 PUFA supplementation during pregnancy. The
supplement recommended in the PACT study was equivalent to 1.2g n-3 PUFA per
day, half the dose used their RCT, so it is unclear if the contrasting effects are due to
the dosage, residual confounding or misclassification in our study. It is also worth
noting that the preventive effect reported by Bisgaard et al[9] was not observed in
the other large RCT[8] or meta-analyses[10] and needs to be confirmed in future
studies.

Reviewer 3 Report

The authors report protective effect of fish consumption in the first year of life on asthma, wheeze and the allergy outcomes at age 6 years. One of the strengths of the study include considering maternal and infant fish or cod liver oil consumption within the same study as well as relevance of fish consumption at an early age. I like different sensitivity analyses to check the robustness of results. There are some potential deficiencies or concerns but the analysis is good and well written, is of importance in the field and should be considered for publication once the responses have been adequately addressed.

Major comments:

Line 89 is not clear: all pregnant women and children aged 6 weeks, 1 year, 2 years…. It is not clear when the pregnant subjects joined the trial and when they started fish oil supplementation. Was it at 6 weeks? Were they monitored and followed up even in second or third trimester? or directly at year 1? This could probably be relevant for those infants who have developed symptoms of eczema or asthma before 6 months. In line 116, do the authors also have data on 2nd and 3rd trimester and whether in addition to 6 weeks, other time points were followed-up? There are papers suggesting the importance of consumption of fish oil in the third trimester (PMID: 19173020) and also as highlighted in the papers below. Since timing is of importance, this could be a reason to not see an association between allergy related outcomes and fish or cod liver oil consumption by mothers. Again, this could be a study design issue, however would be good to know/discuss what authors think about it. Also, while authors mention it as a strength that they follow women during pregnancy, however are there any data demonstrating that supplement intake reported once during pregnancy is representative of intake throughout pregnancy? Some number of FFQs were used to assess fish oil supplementation, but do the authors have actual blood samples on these women at enrollment or throughout, so why not measure the PUFAs directly like EPA and DHA and get some assessment of the accuracy of the FFQ on at least a sample of the population if they have any association with the outcome? Did the authors have information on baseline PUFAs and after supplementation? This would be important to know considering that people from nordic countries in general eat lot of oily fish and therefore could have higher PUFAs than rest of the population. Do the authors have measures on baseline Vitamin D? Understanding that Vitamin D could influence the results, did you adjust for it or checked for interaction? Considering that it is a well-powered population based intervention including both mothers and children, it would be interesting to see if there is any interaction between Vitamin D and fish oil/PUFAs? Since the results could differ between ethnicities and their baseline values, are all the women from the same ethnicity? The title says: Fish consumption at 1 year of age reduces the risk of 2 eczema, asthma and wheeze at 6 years of age but does not cite enough references from recently published literature in the same field. One of the most famous papers published in 2016 on fish oil is not cited: Hans Bisgaard et al., N Engl J Med. 2016 (PMID: 28029926). The authors identified supplementation with n-3 LCPUFA in the third trimester of pregnancy reduced the absolute risk of persistent wheeze or asthma and infections of the lower respiratory tract in offspring by one third. In another recently published paper by Kachroo P et al., J Allergy Clin Immunol Pract. 2019 (PMID: 31226448) using the Vitamin D Antenatal Asthma Reduction Trial (VDAART) food frequency questionnaire data in both first and third trimester, authors identified that the risk of asthma/recurrent wheeze was significantly lower among children of mothers who took prenatal fish oil (14% versus 28%, HR=0.45; one-sided P=0.018) and protective effect was driven by highest doses and for mothers taking fish oil in both first and third trimester. In Lee-Sarwar K et al., J Allergy Clin Immunol Pract. 2019 (PMID: 30145365); For both dietary and plasma measures of total, omega-3, and omega-6 PUFAs, inverse associations with asthma and/or recurrent wheeze and allergic sensitization were strongest among subjects with both high umbilical cord blood 25-hydroxyvitamin D and high PUFA at age 3 years.

Minor comments:

Grammatical errors and spacing should be checked throughout, for eg:

Line 36-37 should be reworded as: Plausible biological mechanisms can explain a causal relationship between increased/increase in dietary N-6 PUFAS and allergy related diseases [4]. Line 52-55 should be reworded as: Few publications have considered maternal and infant fish or cod liver oil consumption within the same study and, due to correlations between maternal and infant dietary habits, there is a need to consider/to study both in order to disentangle their effects or say something like: “there is a need of studies considering both in order to disentangle their effects.” Reference format seems different in line 110 citing REF 26,27. Even though we know, it would be good to add 95% CI (similar to line 249) in other places too like line 251 to be clear what those numbers represent

Author Response

sensitivity analyses to check the robustness of
results. There are some potential deficiencies or concerns but the analysis is good and well written,
is of importance in the field and should be considered for publication once the responses have been
adequately addressed.
Major comments:
Line 89 is not clear: all pregnant women and children aged 6 weeks, 1 year, 2 years…. It is not clear
when the pregnant subjects joined the trial and when they started fish oil supplementation. Was it
at 6 weeks? Were they monitored and followed up even in second or third trimester? or directly at
year 1? This could probably be relevant for those infants who have developed symptoms of eczema
or asthma before 6 months.
Thank you for pointing out that this was not clear. We have made substantial changes to
§2.1. In particular we have now stated clearly that the current analyses do not assess the
effect of the PACT intervention, but aims to identify associations between dietary fish and
cod liver oil using the PACT data. In doing so, the data is essentially analysed as an
observational cohort, pooling data from the “control” and “intervention” cohorts whilst
accounting for cohort membership in the statistical analyses. For this reason, we have chosen
to move the details of the recruitment processes for the control and intervention cohorts to
the Supplementary Materials as we now see these were distracting and confusing.
With regard to monitoring and follow-up: Participants were only followed up in the PACT
study through the lifestyle questionnaires (once during pregnancy then at 6 weeks, 1 year
and 2 years) and the child health questionnaires (at 2 and 6 years). However, routine
antenatal care in Norway includes visits at 8-12 weeks, then 18, 24, 28, 32, 36, 38, 40 and 41,
and women coming to their initial visit after the introduction of the PACT intervention should
have received ongoing encouragement to increase fish intake, and to reduce smoke exposure
and indoor dampness at these checks.
Routine child health follow-ups 6 weeks, and 3, 4, 5, 6, 8, 10, 12, 15 and 18 months, and 2, 4
and 6 years. At each visit, children are assessed by a child health nurse and they are
examined by a doctor at the 6-week, 6-month, 12-month, 2-year and 6-year assessments.
Like the antenatal visits, these not study specific follow-ups, but are routine throughout
Norway and are free of charge. The PACT study did not specifically instruct the nurses and
doctors at child health centers to record signs and symptoms of allergic disease, however the
frequency of these follow-ups and prevalence of allergic diseases means that most infants
who developed signs and symptoms before 6 months or 1 year of age should have been
identified by the health system. However, the parents were not asked to complete any child
health questionnaires before 2 years of age, so there is a risk of recall bias for debut age and
diagnosis of allergic rhinoconjunctivitis. This is now mentioned in the discussion (lines 465-
472 in the clean manuscript):
Allergic outcomes were parentally reported at 2 and 6 years. Whilst there was no
specific follow-up of health outcomes among PACT participants before 2 years of
age, we believe that most children with allergic symptoms would have been
identified through routine follow-ups at community health centers. Throughout
Norway, children attend free-of-charge routine follow-ups with a child health nurse
at 6 weeks, and 3, 4, 5, 6, 8, 10, 12, 15, 18 and 24 months. Children are additionally
assessed by a doctor at 6 weeks, and 6, 12 and 24 months. Nonetheless, the age of
disease or symptom debut recorded in the PACT study may still be subject to recall
bias.
In line 116, do the authors also have data on 2nd and 3rd trimester and whether in addition to 6
weeks, other time points were followed-up? There are papers suggesting the importance of
consumption of fish oil in the third trimester (PMID: 19173020) and also as highlighted in the papers
below. Since timing is of importance, this could be a reason to not see an association between
allergy related outcomes and fish or cod liver oil consumption by mothers. Again, this could be a
study design issue, however would be good to know/discuss what authors think about it.
As noted above, the women only completed the lifestyle questionnaire once during
pregnancy. In Norway, the first routine antenatal check is at around 12 weeks gestation.
However, many women choose to attend early. In the presented analyses, approximately 44
% of women completed the pregnancy questionnaire in the first trimester, 44 % in the second
and the remaining 12 % in the third trimester. The median gestation at completion of the
pregnancy questionnaire was 13 weeks (IQR 10 – 21 weeks) [lines: 233-234]. Since we only
have dietary information at one timepoint in pregnancy, we cannot reliably determine
whether early or later fish consumption has a greater effect on allergic outcomes. We now
mentioned this in the discussion (lines: 459-465).
Only one questionnaire was completed during pregnancy and we therefore do not
have the possibility to determine how representative the reported dietary habits are
of the entire pregnancy period. Nor can we use this data to investigate if importance
of the timing of fish or cod liver oil consumption during pregnancy. Analyses from
the VDAART study suggest that consumption during both the first and third trimester
off the greatest preventive effect against asthma / recurrent wheeze [36], however
this is in a population with comparatively very low fish oil consumption.
Also, while authors mention it as a strength that they follow women during pregnancy, however are
there any data demonstrating that supplement intake reported once during pregnancy is
representative of intake throughout pregnancy?
This is a good point which we have now included in the Discussion section (see above). We
did not find any article specifically reporting the representativeness of one FFQ.
Some number of FFQs were used to assess fish oil supplementation, but do the authors have actual
blood samples on these women at enrollment or throughout, so why not measure the PUFAs directly
like EPA and DHA and get some assessment of the accuracy of the FFQ on at least a sample of the
population if they have any association with the outcome?
We do have plasma samples available for a subset of women and infants who participated in
two sub-studies, however these samples have not been analyzed for PUFAs and we do not
have the capacity to do so in the near future.
Did the authors have information on baseline PUFAs and after supplementation? This would be
important to know considering that people from nordic countries in general eat lot of oily fish and
therefore could have higher PUFAs than rest of the population.
The traditionally higher consumption of oily fish and cod liver oil in Nordic countries would
certainly mean that the baseline PUFA levels could be higher than other populations. These
would be interesting analyses and would help us understand if the PACT intervention further
increased PUFA levels.
However, rather than assessing the intervention, we are simply comparing allergic outcomes
based on reported diet in the current analyses. In this sense, the data is treated as an
observational cohort and we view baseline vs post supplementation questions are not
testable / less relevant. We see that this was not clearly stated in the previous version of the
manuscript as described above.
For completeness, we do not have the possibility to determine baseline PUFA levels and postintervention
PUFA levels for the participating women in the intervention cohort as blood was
only collected once late in pregnancy (in a subset of these women).
Do the authors have measures on baseline Vitamin D? Understanding that Vitamin D could influence
the results, did you adjust for it or checked for interaction? Considering that it is a well-powered
population based intervention including both mothers and children, it would be interesting to see if
there is any interaction between Vitamin D and fish oil/PUFAs?
These are very interesting questions! Vitamin D has not been measured these women and
children, and the lifestyle questionnaires did not have sufficient information about sources of
Vitamin D. We are therefore unable to provide any analysis of interactions between vitamin
D and PUFAs. Whilst the main PACT study is a well-powered population based intervention,
we only blood samples from a small subset of these participants. Nonetheless, we will be
considering this in future studies, and we have briefly mentioned potential interactions
between vitamin D and PUFAs in the discussion (lines 433-442):
In addition to baseline levels of PUFAs, other factors may influence the effect of n-3
PUFAs on allergic outcome, such as vitamin D status or concomitant vitamin D
supplementation. Results from the Vitamin D Antenatal Asthma Reduction Trial
(VDAART) suggest that infants with high cord blood 25(OH)D, in combination with
high plasma or dietary PUFA at 3 years of age, was associated with the lowest odds
of allergic disease[35]. This would suggest that antenatal vitamin D status may
influence the magnitude of the preventive effect of postnatal PUFA consumption.
Interaction between maternal vitamin D and PUFA supplementation was not
observed in by Bisgaard et al [9]. In a separate analysis from the VDAART study,
prenatal fish oil supplementation was associated with a reduced risk of asthma /
recurrent wheeze in offspring, however baseline vitamin D in mothers did not affect
this preventive effect [36].
Since the results could differ between ethnicities and their baseline values, are all the women from
the same ethnicity?
We do not have information about the ethnicities of women included in the study. According
to the Norwegian Bureau of Statistics, 2.8-5.6 % of the population in Trondheim had parents
who were not born in Europe, North America or Oceania during the study period (2000 –
2010). We therefore do not believe that ethnic difference have substantially influenced our
results.
The title says: Fish consumption at 1 year of age reduces the risk of 2 eczema, asthma and wheeze at
6 years of age but does not cite enough references from recently published literature in the same
field. One of the most famous papers published in 2016 on fish oil is not cited: Hans Bisgaard et al., N
Engl J Med. 2016 (PMID: 28029926). The authors identified supplementation with n-3 LCPUFA in the
third trimester of pregnancy reduced the absolute risk of persistent wheeze or asthma and
infections of the lower respiratory tract in offspring by one third. In another recently published
paper by Kachroo P et al., J Allergy Clin Immunol Pract. 2019 (PMID: 31226448) using the Vitamin D
Antenatal Asthma Reduction Trial (VDAART) food frequency questionnaire data in both first and
third trimester, authors identified that the risk of asthma/recurrent wheeze was significantly lower
among children of mothers who took prenatal fish oil (14% versus 28%, HR=0.45; one-sided P=0.018)
and protective effect was driven by highest doses and for mothers taking fish oil in both first and
third trimester. In Lee-Sarwar K et al., J Allergy Clin Immunol Pract. 2019 (PMID: 30145365); For both
dietary and plasma measures of total, omega-3, and omega-6 PUFAs, inverse associations with
asthma and/or recurrent wheeze and allergic sensitization were strongest among subjects with both
high umbilical cord blood 25-hydroxyvitamin D and high PUFA at age 3 years.
Thank you for these suggestions, we have included these references in the introduction and
discussion.
Minor comments:
Grammatical errors and spacing should be checked throughout, for eg:
Line 36-37 should be reworded as: Plausible biological mechanisms can explain a causal relationship
between increased/increase in dietary N-6 PUFAS and allergy related diseases [4].
Changed to “…. can explain a causal relationship between increased dietary N-6 PUFAs…”
Line 52-55 should be reworded as: Few publications have considered maternal and infant fish or cod
liver oil consumption within the same study and, due to correlations between maternal and infant
dietary habits, there is a need to consider/to study both in order to disentangle their effects or say
something like: “there is a need of studies considering both in order to disentangle their effects.”
Thanks for the suggestion, we used this formation in the revised text.
Reference format seems different in line 110 citing REF 26,27.
References now corrected
Even though we know, it would be good to add 95% CI (similar to line 249) in other places too like
line 251 to be clear what those numbers represent
95 % CI included as suggested.

Round 2

Reviewer 1 Report

The authors present in the manuscript results about the effect that fish consumption during the first two years of life, along with the fish consumption of the mother during pregnancy and breastfeeding has on the development of allergic pathology at six years of age. The authors handled a large amount of data and demonstrated experience both on the subject of the manuscript and in the analysis of large quantities of results.

Reviewer 3 Report

Thank you for the extensive revisions of this manuscript!

The authors have responded to and addressed the queries in a comprehensive manner and have made appropriate changes to the manuscript! I am satisfied with their responses and have no further queries or comments.